# A Sensitive Pyrimethanil Sensor Based on Electrospun TiC/C Film

**DOI:** 10.3390/s19071531

**Published:** 2019-03-29

**Authors:** Ling Sui, Tingting Wu, Lijuan Liu, Honghong Wang, Qingqing Wang, Haoqing Hou, Qiaohui Guo

**Affiliations:** Department of Chemistry and Chemical Engineering, Jiangxi Normal University, Nanchang 330022, China; 15079058173@163.com (L.S.); wtt1874983756@163.com (T.W.); 15979385750@163.com (L.L.); WHHong816@163.com (H.W.); Wyuan0816@yeah.net (Q.W.); zxp1014@126.com (H.H.)

**Keywords:** electrospinning, TiC, pyrimethanil, electrochemical sensor

## Abstract

Titanium carbide (TiC) is a very significant transition metal carbide that displays excellent stability and electrical conductivity. The electrocatalytic activity of TiC is similar to noble metals but is much less expensive. Herein, carbon nanofibers (CNFs)-supported TiC nanoparticles (NPs) film (TiC/C) is prepared by electrospinning and carbothermal processes. Well-dispersed TiC NPs are embedded tightly into the CNFs frameworks. The electrochemical oxidation of pyrimethanil (PMT) at the TiC/C-modified electrode displays enhanced redox properties, and the electrode surface is controlled simultaneously both by diffusion and adsorption processes. When TiC/C is applied for PMT determination, the as-fabricated sensor shows good sensing performance, displaying a wide linear range (0.1–600 μM, R^2^ = 0.998), low detection limit (33 nM, S/N = 3), and good reproducibility with satisfied anti-interference ability. In addition, TiC/C shows long-term stability and good application in natural samples. The facile synthetic method with good sensing performance makes TiC/C promising as novel electrode materials to fabricate efficient sensors.

## 1. Introduction

To control plant pests and other diseases, the usage of pesticides is widely relied upon [1]. Some of these pesticides are used for protecting food crops during cultivation and post-harvest storage. However, most of these chemicals may exist in the environment after their usage [2]. Therefore, residues of these pesticides would be discovered in fruits, vegetables, or water, which threaten human health. As one kind of aniline–epyrimidine fungicides, pyrimethanil (PMT, chemical structure shown in Scheme 1) plays a crucial role in agriculture, as it is applied to control leaf scab and some post-harvest diseases [3]. However, the wide usage of PMT would produce residues that may become a possible threat for health and environment. Due to the toxicity of PMT, the European Union Directive on drinking water quality established a maximum allowed concentration (0.1 μg/L) for PMT [4]. Therefore, developing a sensitive method to determine PMT is very important. Recently, PMT residues are monitored commonly by gas chromatography or high-performance liquid chromatography (HPLC) coupled with selective detectors [5,6,7,8]. These technologies display satisfactory accuracy, high reproducibility and reliability, but they are also subject to some inevitable shortcomings, such as the need for trained personnel, expensive instruments, and complex extraction steps. On the other hand, electrochemical methods have attracted much attention for the quantitative analysis of pesticide residues due to their good stability, high sensitivity, and low cost [9,10,11].

Recently, transition metal carbides, such as TaC, TiC, and Mo_2_C, have attracted growing attention due to their high catalytic activity. Additionally, the catalytic activity of these transition metal carbides is similar to that of noble metals, but they are less expensive. Particularly, as a vital transition metal carbide, TiC displays wide applications in supercapacitors [12], dye-sensitized solar cells [13], and lithium-ion batteries [14], owing to its good chemical and thermal stability, high catalytic activity, and low conductivity (6.8 × 10^−5^ Ω/cm). Recently, various TiC nanostructures, such as core–shell [12,15], nanowire [14], nanorod [16], and nanoparticle (NP) [17,18], have been widely exploited. Although TiC NPs can effectively improve its electrochemical activity, they tend to agglomerate, decreasing its specific surface area.

Herein, a hybrid of TiC NPs loaded carbon nanofibers (TiC/C) was synthesized via a simple electrospinning and carbothermal approach. Well-dispersed NPs were dispersed tightly on the skeleton of carbon nanofibers (CNFs). As expected, the TiC/C hybrid displayed superior sensing performance for PMT determination. In addition, TiC/C was also used for the detection of PMT in natural samples.

## 2. Experimental

### 2.1. Reagents

Titanium tetrachloride (TiCl_4_), polyacrylonitrile (PAN), and polyvinylpyrrolidone (PVP, M*_w_* = 1,500,000) were obtained from Sigma-Aldrich. PMT was purchased from Bepharm Co. Ltd. (Guangzhou, China). Phosphate buffer saline (PBS) was prepared by mixing 0.1 M NaH_2_PO_4_ and Na_2_HPO_4_. All of the solutions were dissolved with deionized water, obtained from a Milli-Q water purifying system (18 MΩ cm^−1^). The water samples were obtained from a stream in Nanchang City (Jiangxi, China) and filtered with 0.50 μm nylon. Two kinds of food (apple and cucumber) were obtained from local supermarket. A part of apple and cucumber pericarp was cut up and added into beaker, and some ethanol was added to dissolve the pesticide residues. Assisting ultrasonic treatment was conducted for about 60 min, then the solution was filtrated for further experiments.

### 2.2. Apparatus

ATESCAN VEGA-3 scanning electron microscope (SEM) and a Tecnai G20 transmission electron microscope (TEM) were used to characterize the morphology of TiC/C. Raman spectroscopy (WITec-CRM200 Raman system with a laser wavelength of 532 nm) was used to characterize the microstructure of TiC/C. Thermogravimetric analysis (TGA) was performed on a SDT Q700 thermal analyzer (TA Instruments Co., Tokyo, Japan) under air atmosphere. Electrochemical experiments were tested on CHI 760 E electrochemical workstation (Shanghai, China). A three-electrode configuration was employed to performed electrochemical experiments, of which a platinum wire was used for auxiliary electrode. The reference electrode was Ag/AgCl (saturated KCl), and the bare glass carbon electrode (GC) was used as working electrode. Electrochemical impedance spectroscopy (EIS) was measured in 0.1 M KCl solution containing 5 mM Fe(CN)_6_^3−/4−^ (1:1). The frequency was 1.0 × 10^−2^ ~ 1.0 × 10^5^ Hz.

### 2.3. Preparation of TiC/C

TiC/C hybrid was synthesized by combination of electrospinning and carbothermal processes [19]. Briefly, TiCl_4_ (12 wt% relative to TiCl_4_) was dissolved in PVP (TiCl_4_/PVP). PAN (18 wt% relative to PAN) was dissolved in DMAC (PAN/DMAC), then PAN/DMAC solution was mixed with TiCl_4_/PVP (1:1) solution with continuous string for 4 h at 60 °C. The electrospinning process was under a 30 kV voltage, and the nanofibers were collected onto a rotating drum. The as-electrospun nanofibers were oxidized at 230 °C (2 °C/min, 5 h, air atmosphere), and further thermal treatment at 1000 °C (10 °C/min, 1 h, vacuum). For a comparison, CNFs film (the electrospinning solution was PAN/DMAC without the addition of TiCl_4_/PVP solution) was also prepared under the same condition.

### 2.4. Electrode Preparation

Glass carbon electrode (GC, θ = 3 mm) was polished carefully using Al_2_O_3_ powder. The electrode was rinsed and sonicated twice with distilled water and used for further experiments. An amount of 8 mg/mL of TiC/C was dispersed in a solvent mixture containing 25 μL of Nafion (5 wt%) and 250 μL of distilled water by sonication. Immediately after dispersion, 6 μL of TiC/C slurry was coated onto GC electrode surface (TiC/C/GC). The 0.1 M PBS solution was prepared by purging with high purity nitrogen (N_2_) for 30 min and a N_2_ blanket was maintained above the solution during measurement.

## 3. Results and Discussion

### 3.1. Characterization

TiC/C hybrid was synthesized from PAN/TiCl_4_ nanofibers. During the carbothermal process, the nanofibers were carbonized into CNFs, and TiC NPs were formed in situ. Therefore, TiC/C hybrid was incorporated into one step. It can be seen from Figure 1A that TiC/C revealed 3D network structure, and TiC NPs were homogeneously dispersed on the surface of CNFs. The detailed morphology of TiC/C was further performed by TEM technique. The results showed that NPs were embedded into the CNF frameworks (Figure 1B).

The electrical conductivity of TiC/C film was evaluated via a four-point probe method using the following equation [19]:(1)$$\sigma =\frac{L}{RA}$$
where L is the distance between two electrode (cm), R is the resistance of TiC/C film (Ω), and A is the cross-sectional area of TiC/C film (cm^2^). The electrical conductivity of CNFs was 0.50 S·cm^−1^. However, the electrical conductivity of TiC/C increased to 25.5 S·cm^−1^. The results revealed that TiC NPs embedded CNFs could enhance the percolation-type of conduction [20].

Raman spectroscopy is widely used to investigate the structure of molecule and crystal lattice. There were two strong peaks centered at 1338 and 1575 cm^−1^, which were the graphite peaks of TiC/C, and other peaks at 264, 416, and 602 cm^−1^ corresponded to the TiC phase (JCPDS: 65–0242) (Figure 2A), confirming the formation of TiC nanocrystal. TGA analysis was used to measure the thermal stability of TiC/C. As shown in Figure 2B, a weight increase (250–550 °C) was obtained from TiC/C. The increase of weight was due to TiC oxidation into titanium dioxide (TiO_2_) in air atmosphere (TiC + 2O_2_→TiO_2_ + CO_2_) [19]. The amount of TiC was 40.3% in TiC/C, according to the change of weight [21]. Additionally, the temperature of 5% weight loss in TiC/C was 550 °C, which was higher than that of CNFs (390 °C), demonstrating that TiC/C owned much better thermal stability.

### 3.2. Electrochemical Behavior

The electrochemical behaviors of PMT at TiC/C were systematically investigated. As shown in Figure 3A, there was a weak anodic peak centered at 1.13 V at bare GC. A strong anodic peak at 1.08 V could be seen at the CNFs-modified electrode. The peak current of CNFs increased compared with that of bare GC, attributing to the 3D network structure of CNFs, which increased the active area of the electrode. With regard to TiC/C, the anodic peak potential of PMT was observed at a lower potential (1.02 V), and the anodic peak current was almost twice compared to that of CNFs, suggesting that the electrocatalytic activity of TiC/C hybrid was improved with the introduction of TiC NPs. Meanwhile, no cathodic peak appeared in the reverse scan, indicating that the electrochemical process of PMT was irreversible. The results suggested that TiC/C might be employed as a PMT electrochemical probe with high catalytic activity.

The electrochemical impedance spectroscopy (EIS) was performed to analyze the surface properties of electrode. The electron transfer resistance (R_ct_) values were 1.56 × 10^3^, 1.23 × 10^3^, and 5.88 × 10^3^ Ω/cm^2^ for CNFs, TiC/C, and bare GC, respectively (Figure 3B). When CNFs was modified on the bare GC, the value of Rct decreased, indicating that CNFs formed a fast electron transfer pathway. In addition, the Rct of TiC/C was lower than that of the CNFs-modified electrode, indicating that TiC NPs could facilitate the electron transfer between TiC/C film and electrode surface.

### 3.3. Effect of pH

The effect of pH was investigated by CV. The peak current intensity increased from 2.0 to 4.0 and decreased when the pH value further increased (Figure 4A). Thus, pH 4.0 was selected as an optimal value. The plot of *E*_pa_ verus pH was linear, the regression equation was: *E*_pa_ (mV) = −56/Ph + 1.32 (R = 0.998) (Figure 4B). The obtained slope value (56 mV/pH) was close to the theory value (59 mV/pH), suggesting that identical numbers of protons and electrons participated in the reaction.

### 3.4. Effect of Scan Rate

To assess the kinetic process, different scan rates for PMT oxidation were investigated. The peak potential shifted to positive when increased the scan rate (Figure 5A), suggesting that the electrochemical process was a kinetic limitation, and the current increased linearly with the square root of scan rate. The linear regression equation was: *I*_p_ (μA) = 5 + 115.6 ν^1/2^ (V/s, R = 0.999) (Figure 5B), suggesting that a diffusion-controlled process had taken place at TiC/C. Also, the plot of *log I_p_* versus *log* v was found to be linear (50–250 mV/s): *log I*_p_ = 1.42 + 0.62 *log* ν (R=0.993). The slope (0.62) was higher than the theoretical value (0.5), suggesting that this reaction was a diffusion-controlled process. However, the value was less than the theoretical value of 1.0 (adsorption-controlled process), illustrating that the oxidation of PMT at TiC/C also was an adsorption-controlled process [22,23]. The above results suggested that the oxidation of PMT at TiC/C was dominated by diffusion process accompanied by adsorption process [24].

For an irreversible reaction, the peak potential (*E*_p_) is described as follows [25]:(2)$$E_{p}=E^{0}+\left(\frac{RT}{\alpha nF}\right)\ln\left(\frac{RTk^{0}}{\alpha nF}\right)+\frac{RT}{\alpha nF}\ln\upsilon$$

The plot of *E*_p_ versus *ln* ν was linear, and its equation was: *E*_p_ (V) = 1.18 + 0.06 *ln* ν (R = 0.992) (Figure 5C). The calculated value of αn was 0.97. The electron number could be obtained from the Tafel curve. As shown in Figure 5D, the *E*_p_ was proportional to *log* ν, and could be described by the following equation: *E*_p_ (V) = 1.18 + 0.056 *log* ν (R = 0.996). The Tafel slope of b was obtained from the following equation [26,27]:
(3)$$E_{p}=\frac{b\left(log\;\upsilon \right)}{2}+C$$
where
(4)$$\mathrm{Slope}=b=\frac{2.303RT}{\left(1-\alpha \right)nF}$$
The Tafel slope of b was calculated as 0.103. Therefore, the calculated electron transfer number (*n*) was 0.97 (close to 1). Thus, it could be inferred that one proton and one electron took part in the electrochemical reaction.

### 3.5. Determination of PMT by DPV

Figure 6A was the CVs of TiC/C with PMT ranging from 10 to 30 μM. The peak current increased when increasing PMT concentration. This result denoted that the effectively electrochemical sensing ability of TiC/C without fouling effect. As a sensitive electrochemical technique, DPV was selected for PMT determination. As expected, the peak current of PMT increased linearly with its concentration (Figure 6B). The linear range was 0.1–600 μM (R^2^ = 0.998). The detection limit was 33 nM (S/N = 3). The linear equation was: *I* (μA) = 0.72 + 0.02 C_PMT_ (μM) (R^2^ = 0.998) (Figure 6C). The sensing performances of PMT sensor were compared to other methods/materials (Table 1). The results displayed that the TiC/C exhibited a wider linear range or lower detection limit compared to the reported results [28,29,30,31], suggesting TiC/C was a promising material for PMT detection.

### 3.6. Reproducibility, Stability, and Interference Study

The reproducibility of the as-fabricated sensor was also evaluated. The relative standard deviation (RSD) for six electrodes was 3.4%. The RSD of each electrode for six measurements was 2.35%. To assess the stability of TiC/C, a day-to-day stability was investigated. Satisfied stability was found, and the RSD was less than 4.7%, demonstrating that the sensor possessed good reproducibility and stability. The acceptable stability of TiC/C can be due to the embedded TiC NPs, which could prevent NPs detachment and agglomeration from TiC/C frameworks. 

The anti-interference ability is an important parameter for PMT detection. Some common compounds, such as NO_3_^−^, Mg^2+^, SO_4_^2−^, Zn^2+^, Ca^2+^, Na^+^, K^+^, PO_4_^3−^, glucose, vitamin C, and thiabendazole, may coexist with PMT in the natural samples. Under the optimized conditions, the oxidization peak current of 50 μM PMT was individually measured in the presence of different interferents and their influence on the analytical signal was investigated. The results indicated that 50-fold concentration of thiabendazole, 100-fold concentrations of glucose and vitamin C, and 500-fold concentrations of NO_3_^−^, Mg^2+^, SO_4_^2−^, Zn^2+^, Ca^2+^, Na^+^, K^+^, and PO_4_^3−^ showed negligible influence on the PMT (current signal change < 5%) (Figure 6D), indicating a good anti-interference ability of TiC/C.

### 3.7. Natural Sample Analyses

To evaluate the applicability of TiC/C sensor, water, apple, and cucumber were selected for quantitative analysis. Satisfactory recoveries (96.50–101.07%) and low RSDs (2.5–3.8%) were found (Table 2), indicating the acceptable reliability and applicability of TiC/C in analytical applications. 

## 4. Conclusions

TiC/C hybrid was fabricated for the determination of PMT. Homogeneous TiC NPs embedded into the CNFs framework was observed, which integrated the large surface area and unique 3D-networked structure of CNF with the good electrocatalytic activity of TiC NPs. The TiC/C-modified electrode showed an enhanced electrochemical response for the oxidation of PMT. Wide linear range (0.1–600 μM, R^2^ = 0.998) and low detection limit (33 nM, S/N = 3) were obtained. In addition, TiC/C also showed long-term sensing stability and reproducibility. The TiC/C showed potential applications in the fabrication of electrochemical sensors.

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
