# Peer review of "A Sensitive Pyrimethanil Sensor Based on Electrospun TiC/C Film"

_sensors, 2019, doi:10.3390/s19071531_

Round 1

Reviewer 1 Report

The authors are addressing one of the most current pesticide-induced problems. The paper needs to be 

The experimental procedures must contain more detail for the readers to reproduce the work later. For example, the phrase "PAN/DMF (18 wt%) was mixed with PVP/TiCl4 solution" does not contain the details of the proportions each solution in the mixture. The preparation section needs to be rewritten to make it more clear and understandable to future readers. In a similar way, the electrode preparation is not clear and this section needs to be rewritten

Proper experimental control is missing in the SEM and Raman scattering. The authors have shown the TGA of CNF and the composite while the SEM and XRD of CNF is missing. It is important that the authors provide this information for the readers.

As the authors themselves have highlighted that the interference of assay is a very important parameter in the evaluation of the technique, The details of the experimental conditions used in interference study are missing. Was the concentration of PMT kept constant in all these interference studies? The authors may present the detail in the experimental section or the results and discussion as they may wish. But this data is very critical for any researcher to understand and appreciate this study. 

Can the authors shed light on more details like how much surface area is lost due to deposition of TiC particles on the surface of the CNFs? As the particles may be half or partly embedded inside the CNFs and partly be exposed to the environment I suggest the authors compare the surface area of naked TiC particles to the same amount of TiC nanoparticles embedded in CNFs. 

The manuscript needs to be extensively revised to be publishable which include English corrections

Author Response

Thank you very much for giving us very valuable suggestions. We have considered your questions seriously, and answered them as below.

Q1: The experimental procedures must contain more detail for the readers to reproduce the work later. For example, the phrase "PAN/DMF (18 wt%) was mixed with PVP/TiCl4 solution" does not contain the details of the proportions each solution in the mixture. The preparation section needs to be rewritten to make it more clear and understandable to future readers. In a similar way, the electrode preparation is not clear and this section needs to be rewritten.

R: Thank you for your helpful suggestion. The contents of material preparation and electrode preparation have been modified now (Page 5 and 6): "2.3 Preparation of TiC/C: TiC/C hybrid was synthesized by combination of electrospinning and carbothermal processes [19]. Briefly, TiCl4 (12 wt.% relative to TiCl4) was dissolved in N,N-dimethyl acetamide (DMAC). PAN (18 wt.% relative to PAN) was dissolved  in DMAC, then PAN/DMAC solution was mixed with PVP/TiCl4 (1:1) solution with continuous string for 4 h at 60 °C. The electrospinning process was under a 30 kV voltage, the nanofibers were collected onto a rotating drum. The as-electrospun nanofibers were oxidized at 230 °C (2 °C/min, 5 h, air atomsphere), and further thermal treatment at 1000 °C (10 °C/min, 1 h, vacuum). For a comparison, CNFs film was also prepared under the same condition. 2.4 Electrode preparation: Glass carbon electrode (GC, Ɵ = 3 mm) was polished carefully using Al2O3 powder. The electrode was rinsed and sonicated twice with distilled water and used for further experiments.. 8 mg/mL of TiC/C was dispersed in a solvent mixture containing 25 μL of Nafion (5 wt.%) and 250 μL of distilled water by sonication. Immediately after dispersion, 6 μL of TiC/C slurry was coated onto GC electrode surface (TiC/C/GC). The 0.1 M PBS solution was prepared by purging with high purity nitrogen (N2) for 30 min and maintained a N2 blanket above the solution during a measurement."

Q2: Proper experimental control is missing in the SEM and Raman scattering. The authors have shown the TGA of CNF and the composite while the SEM and Raman scattering of CNF is missing. It is important that the authors provide this information for the readers.

R: Thank you for your helpful suggestion. Control experiments of SEM (Page 6) and Raman scattering (Page 7) of CNF have been added in the manuscript:

Figure 1. SEM (A) and TEM (B) images of TiC/C; Inset was the SEM image of CNF.

Figure 2. Raman spectra (A) and TGA curves (B) of TiC/C and CNF in air atmosphere.

Q3: As the authors themselves have highlighted that the interference of assay is a very important parameter in the evaluation of the technique, The details of the experimental conditions used in interference study are missing. Was the concentration of PMT kept constant in all these interference studies? The authors may present the detail in the experimental section or the results and discussion as they may wish. But this data is very critical for any researcher to understand and appreciate this study. 

R: Thank you for your helpful suggestion. The details of the experimental conditions used in interference study have been added in the results and discussion section (Page 14)"...Under the optimized conditions, the oxidization peak current of 50 μM PMT was individually measured in the presence of different interferents and their influence on the analytical signal was investigated. ...".

Q4: Can the authors shed light on more details like how much surface area is lost due to deposition of TiC particles on the surface of the CNFs? As the particles may be half or partly embedded inside the CNFs and partly be exposed to the environment I suggest the authors compare the surface area of naked TiC particles to the same amount of TiC nanoparticles embedded in CNFs. 

R: Thank you for your helpful suggestion. Although we want to shed light on more details like how much surface area is lost due to deposition of TiC particles on the surface of the CNFs, we could not obtain TiC particles from our work since CNF substrate and TiC particles were formed simultaneously.

Q5: The manuscript needs to be extensively revised to be publishable which include English corrections.

R: Thank you for your helpful suggestion. English including English corrections has been revised carefully throughout the manuscript now.

Reviewer 2 Report

The manuscript is about construction of sensor for pyrimethanil (PMT) based on TiC/C film obtained using electrospinning. The sensor approach is interesting and the obtained results are clearly presented graphically. However, their description is too technical and it needs more discussion on mechanisms.

General comments:

English needs significant revision

Chemical structure of PMT would be very useful.

More convenient presentation of EIS spectra is when they are normalised per geometric area. It is easier to compare with other works in this way.P 5

Detailed comments:

Film description in the title is incorrect, it should be TiC/C.

Abbreviation of CNF is not explained in the text and, therefore, it is not clear which material      was used for control.

Fig. 3B: please “square” the EIS spectra, it means on the both axes should be the same interval per unit because the same units are on both of them. Now spectra are misshapes and do not present real view.

Page 4 after Fig. 3: the authors state that PMT was done on Ti/CNF. Is it correct?

Page 5: From the data it is clear that dominating is diffusion and these experiments are performed to determine the limiting step. Adsorption and diffusion as well as electron transfer are always occurring in electrochemical reactions.

Calibration plot looks like it has two linear ranges. The first one is at a low concentrations.      Isn’t it so?

What kind of natural samples the authors mean in Section 3.6? Please change “practical”      to “natural”.

Conclusions are too general. They need some numbers to confirm these statements.

Author Response

Q1: English needs significant revision.

R: Thank you for your helpful suggestion. English has been revised carefully throughout the manuscript now.

Q2: Chemical structure of PMT would be very useful.

R: Thank you for your helpful suggestion. The Chemical structure of PMT has been added in the text (Page 12).

Scheme 1. The Chemical structure of PMT.

Q3: More convenient presentation of EIS spectra is when they are normalised per geometric area. It is easier to compare with other works in this way.

R: Thank you for your helpful suggestion. We think the unit of EIS plot is "Ohm, Ω", which is a international unit. Therefore, it is easy to compare with other works without normalize per geometric area.

Q4: Film description in the title is incorrect, it should be TiC/C.

R: Thank you for your helpful suggestion. The title has been modified as: "A sensitive pyrimethanil sensor based on electrospun TiC/C film ".

Q5: Abbreviation of CNF is not explained in the text and, therefore, it is not clear which material was used for control.

R: Thank you for your helpful suggestion. The abbreviation of CNF, which was appeared first time, has been added in the text (Page 4): "carbon nanofibers (CNF)".

Q6: Fig. 3B: please “square” the EIS spectra, it means on the both axes should be the same interval per unit because the same units are on both of them. Now spectra are misshapes and do not present real view.

R: Thank you for your helpful suggestion. The axes of EIS spectra were square now.

Figure 3 (A) CV curves of bare GC, CNFs and TiC/C modified electrodes in 0.1 M PBS (pH 4.0) containing 50 μM PMT; (B) Nyquist plots of bare GC, CNFs and TiC/C modified electrodes.

Q7: Page 4 after Fig. 3: the authors state that PMT was done on Ti/CNF. Is it correct?

R: Thank you for your helpful suggestion. The sentence of "... PMT was done on Ti/CNF ..." was changed to " ...PMT was done on TiC/C...".

Q8: Page 5: From the data it is clear that dominating is diffusion and these experiments are performed to determine the limiting step. Adsorption and diffusion as well as electron transfer are always occurring in electrochemical reactions.

R: Thank you for your helpful suggestion. About the adsorption or diffusion process, we has been modified in the text (Page 11):"...The above results suggested that the oxidation of PMT at TiC/C was dominated by diffusion process accompanied by adsorption process [24]".

Q9: Calibration plot looks like it has two linear ranges. The first one is at a low concentrations. Isn’t it so?

R: Thank you for your helpful suggestion. Almost all of the data were on the calibration plot even at a low concentrations (the R2 was 0.998). Therefore, we think there was only one linear range (0.1-600 μM, R2=0.998).

Q10: What kind of natural samples the authors mean in Section 3.6? Please change “practical” to “natural”.

R: Thank you for your helpful suggestion. In section 3.6, the word of "practical" has been changed to "natural" now.

Q11: Conclusions are too general. They need some numbers to confirm these statements.

R:  Thank you for your helpful suggestion. Conclusions section has been modified (Page 15): "TiC/C hybrid was fabricated for the determination of PMT. Homogeneous TiC NPs embedded into the CNFs framework was observed, which integrated the large surface area and unique 3D-networked structure of CNF with the good electrocatalytic activity of TiC NPs. TiC/C modified electrode showed enhanced electrochemical response for the oxidation of PMT. Wide linear range (0.1–600 μM, R2=0.998) and low detection limit (33 nM, S/N=3) were obtained. In addition, TiC/C also showed long-term sensing stability and reproducibility. The TiC/C showed potential applications in the fabrication of electrochemical sensors. "

Round 2

Reviewer 1 Report

The paper now includes most of the necessary details. The manuscript may be accepted with minor corrections in texts. 

Author Response

Response to Reviewer 1 Comments

Thank you very much for giving us very valuable suggestions. We have considered your questions seriously, and answered them as below.

Point 1: The paper now includes most of the necessary details. The manuscript may be accepted with minor corrections in texts. 

Response 1: Thank you for your helpful suggestion. English including English corrections has been revised carefully throughout the manuscript now.

For example, In Abstract section:

(1) "conductivity " was changed to "electrical conductivity";

(2) "...in real samples" was changed to "...in natural samples ";

(3) " ...efficent sensors" was changed to " ...efficient sensors ".

2.1. Reagents of Experimental section:

(1) " Titaniumtetrachloride "was changed to "Titanium tetrachloride";

(2) "18 MΩ cm" was changed to "18 MΩ cm1";

Other mistakes and corrections have been revised throughout the manuscript.

Reviewer 2 Report

The manuscript is much better now but some revision is required prior to the publication. The authors made changes in the text not carefully, some places are not marked, some are stated in the replies but not entered to the text.

Comments:

English needs a great revision, because no changes are visible in comparison to the previous version

Spelling mistake in the title.

EIS spectra are not squared

Although Ohm is international unit, but resistance is area depended parameter, therefore,      it is difficult to compare immediately, a reader has to recalculate it normalizing per are to      compare with other articles or their own works.

Please move Scheme 1 to Introduction.

Author Response

Response to Reviewer 2 Comments

Thank you very much for giving us very valuable suggestions. We have considered your questions seriously, and answered them as below.

Point 1: English needs a great revision, because no changes are visible in comparison to the previous version.

Response 1: Thank you for your helpful suggestion. English has been revised carefully throughout the manuscript now. For example, In Abstract section:

(1) "conductivity " was changed to "electrical conductivity";

(2) "...in real samples" was changed to "...in natural samples ";

(3 )" ...efficent sensors" was changed to " ...efficient sensors ".

2.1. Reagents of Experimental section:

(1) " Titaniumtetrachloride "was changed to "Titanium tetrachloride";

(2) "18 MΩ cm" was changed to "18 MΩ cm−1";

Other mistakes and corrections have been revised throughout the manuscript.

Point 2: Spelling mistake in the title.

Response 2: Thank you for your helpful suggestion. The title has been modified as: "A sensitive pyrimethanil sensor based on electrospun TiC/C film ".

Point 3: EIS spectra are not squared.

Response 3: Thank you for your helpful suggestion. The axes of EIS spectra were square now (Page 4).

Figure 3. (A) CV curves of bare GC, CNFs and TiC/C modified electrodes in 0.1 M PBS (pH 4.0) containing 50 μM PMT; (B) Nyquist plots of bare GC, CNFs and TiC/C modified electrodes.

Point 4: Please move Scheme 1 to Introduction.

Response 4: Thank you for your helpful suggestion. Scheme 1 has been moved to Introduction now (Page 2).